# Reducing Young Schoolchildren’s Intake of Sugar-Rich Food and Drinks: Study Protocol and Intervention Design for “Are You Too Sweet?” A Multicomponent 3.5-Month Cluster Randomised Family-Based Intervention Study

**DOI:** 10.3390/ijerph17249580

**Published:** 2020-12-21

**Authors:** Sidse Marie Sidenius Bestle, Bodil Just Christensen, Ellen Trolle, Anja Pia Biltoft-Jensen, Jeppe Matthiessen, Sarah Jegsmark Gibbons, Bjarne Kjær Ersbøll, Anne Dahl Lassen

**Affiliations:** 1Division of Food Technology, National Food Institute, Technical University of Denmark, Kemitorvet Building 201, 2800 Kongens Lyngby, Denmark; boju@food.dtu.dk (B.J.C.); eltr@food.dtu.dk (E.T.); apbj@food.dtu.dk (A.P.B.-J.); jmat@food.dtu.dk (J.M.); sajegi@food.dtu.dk (S.J.G.); adla@food.dtu.dk (A.D.L.); 2Department of Applied Mathematics and Computer Science, Technical University of Denmark, 2800 Kongens Lyngby, Denmark; bker@dtu.dk

**Keywords:** sugar-rich food, discretionary food, SSBs, family-based intervention, social cognitive theory, dietary guidelines

## Abstract

A high consumption of sugar-rich discretionary food and drinks has several health implications, which have been traced from childhood into adulthood. Parents act as primary mediators shaping children’s dietary habits, and interventions that engage parents have shown to result in positive outcomes. Further, collaboration with local school health nurses and dentists provides an effective structural frame to support behaviour change and anchor new initiatives. The multicomponent 3.5-month cluster randomised family-focused intervention “Are you too Sweet?” aims to evaluate the effectiveness of communicating new Danish guidelines for sugar-rich discretionary food and drinks for school starters (5–7 years). This paper describes the development, outcomes and process evaluation of the intervention that includes three main components: extended dialogue during a school health nurse consultation, a box with home-use materials, and a social media platform to facilitate interaction among participants. Children (*n* = 160) and their parents were scheduled for a baseline interview at six different schools. The intervention was developed to increase self-efficacy, knowledge about guidelines, observational learning and reduce impediments for behavioural change. The desired primary outcome was a reduction in intake of sugar-rich food measured through a 7-day dietary record. The results contribute to the evidence on effective health promotion strategies.

## 1. Introduction

A high intake of sugar-rich food and sugar-sweetened beverages (SSB) in children has several health implications. Evidence supports the association between a high intake of SSB and the risk of overweight in children and adults [1,2,3,4,5,6]. A Swedish cross-sectional study among adults suggested that the higher the intake of added sugar in the diet, the more likely it is that the intake of micronutrients will be compromised [7]. Studies from both Denmark and Norway have shown lower intake of dietary fibre, vitamin E, vitamin D, and iron in children with the highest sugar intake compared with children with the lowest intake [8,9]. Likewise data from the UK National Diet and Nutrition Survey 2008–2012 have shown that children with an intake of free sugars above 13% of total energy (E%) had a lower intake of micronutrients compared with children with a lower intake [10]. An Australian study reported that children whose intake of free sugars exceeded 20 E% to be more likely to have inadequate micronutrient intake [11]. Finally, strong evidence supports that sugar intake contributes to the development of dental caries [12,13].

Based on the recognition that the negative health effects, especially dental caries, are cumulative, tracking from childhood to adulthood, the World Health Organization (WHO) recommends reducing the intake of free sugars throughout the life course and lowering the intake of free sugars to less than 10 E%, and preferably to less than 5 E% [14]. The Nordic Nutrition Recommendations (NNR) also advises consuming a diet with less than 10 E% from added sugar [15]. The US Scientific Report of the 2020 Dietary Guidelines suggests a recommendation <6 E%, since they suggests *“[…] that less than 6 percent of energy from added sugars is more consistent with a dietary pattern that is nutritionally adequate, while avoiding excess energy intake from added sugars, than is a pattern with less than 10 percent energy from added sugars.”* [5]. Compared to added sugar, the term free sugar include also sugars naturally present in honey, syrups fruit juice concentrates, which typically is not included in added sugar [5,15,16]

Intake of added sugar mostly comes from “discretionary food”, a term used to define energy-dense but nutrient-poor foods, such as SSBs, pastries, cakes, sweets, chocolate, ice cream, or salty snacks [17]. A Danish qualitative study showed that although parents are well aware of the consequences of eating large amounts of discretionary food, there is great uncertainty as to how much discretionary food is too much [18]. Further, there is great variation between intake of discretionary food during weekdays and weekends, with an intake during weekends exceeding recommendations substantially, i.e., an intake of 14 E% added sugar on average for 4–13 year-old Danish children on Friday and Saturday [19].

Children’s high consumption of discretionary food and drinks [16,20,21,22], with the increased risk of establishing unhealthy habits throughout life, call for sustainable prevention strategies. Several systematic reviews have evaluated the effectiveness of interventions to reduce consumption of discretionary food, especially SSBs, both targeting environmental changes and behavioural changes [13,23,24,25]. Notably, home-based environmental interventions with inclusion of parents have proven to be efficient [23]. However, interventions on reducing sugar-rich discretionary food using theoretical-based family-centred approaches have been limited. Social cognitive theory (SCT) provides a suitable guide for behavioural change in families, with a focus on strengthened self-efficacy through reciprocal determinism, the interaction between humans and environment, and observational learning.

Most European countries have implemented a form of school health service to provide school children with various types of healthcare [26]. In Denmark, children enter the school system into “pre-school class” around the age of six. During this first year, all children together with their parents are invited to a health consultation at the school with the school health nurse. This consultation evolves around communicating both wellbeing and health, and offers an opportunity to communicate guidelines on diet, physical activity, screen time and sleep. The intervention “Are you too Sweet?” aims at using this setting to deliver newly developed guidelines on discretionary food and drinks, and to share several materials developed based on SCT, to engage families to change habits related to discretionary food and drinks. “Are you too Sweet?” involves collaboration with both the school health nurses, the dentist care, and communication partners. This paper describes the development of the components and the study protocol of the multicomponent family-based “real-life” cluster-randomised intervention “Are you to Sweet?” evaluating the feasibility and effectiveness of different strategies for changing the family sugar culture and reducing the intake of sugar-rich discretionary food and drinks among young schoolchildren aged 5–7 years.

## 2. Materials and Methods

### 2.1. The Municipality Setting

In Denmark, primary health prevention in children is part of the local school health nurse’s domain, managed by municipalities. When children are enrolled in the school system, school health nurses consult the families at the school. New national guidelines for schoolchildren, and strategies to communicate these, would have to be adopted at the municipality level. The intervention “Are you too Sweet?” will be performed in the Danish municipality Hvidovre. This municipality is chosen as it represents the national mean for socio-economic status, ethnicity, and education level [27].

### 2.2. Study Design and Setting

“Are you too Sweet?” is a two-arm, parallel, cluster-randomised controlled trial with clusters defined as schools within the municipality. The study investigates the effectiveness of a 3.5-month multicomponent family-based intervention against a control group receiving standard care by the school health nurses. The intervention involves school starters 5–7 years and their parents, in a Danish municipality (Hvidovre), focusing on reducing intake of sugar-rich discretionary food and drinks. Baseline measures will run from October until December 2020, and follow-up measures will run from February until March 2021. Overall, the intervention will consist of a parent performed registration of the child’s intake of discretionary food and drinks, a focused health consultation with the school health nurse, a box with materials and tools to be used at home handed out by the school nurses and finally a social media platform for interaction between participating parents. A flow of the intervention is shown in Figure 1.

#### 2.2.1. Study Population and Sample Size

Eligible children are children starting public pre-school education at six chosen schools in Hvidovre municipality in August 2020. There is no inclusion criteria regarding diet or disease. Power calculations were performed based on 6–7 year-old children from the Danish National Survey of Diet and Physical Activity [28]. Calculations determined that 76 participants will be needed in both intervention and control groups to detect a 25% reduction in the intake of sugar-rich discretionary foods at a power of 80% and a 95% confidence interval. For a 25% reduction in intake of added sugar, 63 participants will be needed. To account for dropouts or loss of participants to follow-up, an enrolment of a minimum of 100 families in the intervention group and an additional 100 families in the control group was planned.

#### 2.2.2. Recruitment and Randomisation

Participants were recruited during the period of March–May 2020. The local dental clinician phoned parents of children starting school in the districts of the six schools in 2020 and asked if they were interested in participating in the intervention. Beforehand, the participants had received the participant information sheet via their personal digital post-box. In total, 237 parents, corresponding to 54% of children starting at the six local schools in Hvidovre, initially accepted to participate. The most common reason for declining participation was lack of time and/or resources. The parents were then contacted again after the start of school, and 160 children and their parents ended up scheduling an appointment at baseline.

Cluster randomisation of schools took place after the recruitment was completed, and children were assigned to the control or intervention group according to randomization of their school. For an even distribution between the control and intervention group, the schools were randomised by the number of children at each school and the schools’ socioeconomic index, using R statistical software package, version 4.0. Information about the individual schools’ socioeconomic index was obtained from the municipality’s school division. For practical reasons, randomisation had to be completed before baseline, and therefore could not be blinded for the investigators. Randomisation was blinded to the participants until after they completed a 7-day dietary record at baseline. An unexpected large dropout rate prior to baseline measures created partly by COVID-19 related disruptions caused concern on the sample size. Thus, based on the initial randomisation parameters (number of children and socioeconomic index), one control school was changed to an intervention school in order to secure statistical power in the intervention group. The distribution between intervention (*n* = 4) and control schools (*n* = 2) is thus unbalanced, however, smaller control groups in public health intervention studies have been shown to be valid previously [29].

#### 2.2.3. Standard Care—Control Group

All children in Hvidovre municipality are offered a consultation with the school health nurse together with their parents when they start pre-school class. The school health nurse dialogues with the child and the parents about health and wellbeing, and individual problems if the child has difficulties in school or at home. In the project “Are you too Sweet?” updated teaching materials, regarding diet, physical activity, screen time, and sleep among 6–15 year-old children, and a conversation tool (Figure 2), illustrating a daily schedule, were provided to the school health nurses, to ensure standardized consultation in the project. The conversation tool was developed on the basis of two focus group discussions with the school health nurses, five days of observations of school health consultations, and a literature search on making conversations with children [30,31]. Focus in the conversation tool is the five main target areas for the conservation: diet, physical activity, screen time, sleep and wellbeing.

### 2.3. Intervention Components and Theoretical Framework

“Are you too Sweet?” includes three main elements aimed for the intervention group. The foundation of the project promotes newly developed guidelines on a maximum amount of sugar-rich discretionary food and drinks. The first element consists of a “sugar-rich food screener” assessing intake of pre-defined discretionary food and drink followed by an extended dialogue during the consultation with the school health nurse. The intervention group receives five extra minutes of consultation time, focusing on intake of sugar-rich discretionary food and drinks based on the new guidelines. The second element consists of a box of home-use materials to engage and inspire the family to decrease their intake of sugar-rich discretionary food and drinks, but at the same time maintain or increase time spent together within the family. The box includes a card game, an inspiration booklet describing different strategies to cut down on discretionary food and drinks, a serving size board with illustrations of the amount of servings for discretionary food and drink choices in a healthy diet, several activity suggestions to inspire local public family activities, and a children’s book. The third element is the communication and social media platform for participants to interact peer-to-peer and receive, e.g., information on positive dietary changes, and at the same time be reminded of participation in the project. An overview of the intervention parts is provided in Table 1. The box with materials given to the intervention group will be given to children in the control group after follow-up measurements.

“Are you too Sweet?” is guided by social cognitive theory (SCT) [32,33]. As the aim is to target the home environment, involving both parents and children, SCT is suitable to operate at an interpersonal level. Concepts from the SCT used are: *self*-*efficacy*, *knowledge* and *behavioural capability*, *expectations*, *observational learning*, *social*/*structural impediments* and *reciprocal determinism* [32,33]. Further, as parents act as a primary mediator in our intervention, concepts drawn from studies on parental practices were used [34] in order to ensure evidence-based parental guidance. Each of the components in the intervention is described in more detail in the following sections.

#### 2.3.1. New Guidelines

As a foundation for the intervention, new guidelines for maximum intake of discretionary food and drinks were developed. We developed a model on nutritional profiling to classify foods regarded as discretionary foods. In this model, foods and drinks were evaluated based on 20 different qualifying nutrients, four disqualifying nutrients, and the energy density. The foods regarded as discretionary foods, calculated from the model consisted of sugar-rich foods (cakes, confectionary, ice cream and chocolate), energy-dense snacks (energy-bars, crackers and chips) and discretionary drinks (SSBs, fruit drink concentrate and artificially sweetened beverages). The amount of energy for discretionary food and drinks choices in a healthy diet was calculated for two age groups, 4–6 years and 7–9 years. For this calculation, the average Danish diet in each age/gender group was modelled to meet nutritional requirements (RI) as well as official Danish dietary guidelines. Thus, the modelled diet is healthier than an average Danish diet. The amount of discretionary food and drinks is communicated as serving sizes corresponding to 450 kJ. The maximum intake advised for 4–6 years, and for 7–9 years of pre-defined discretionary food was four and five servings per week, respectively.

#### 2.3.2. Sugar-Rich Food Screener

The sugar-rich food screener was developed as a web-based dietary registration tool aimed for individual direct feedback, illustrating the share that discretionary food and drinks takes up of daily energy consumption (Figure 3). The sugar-rich food screener takes approximately 10 min to complete. It can be completed on either a mobile phone, tablet or computer, and was developed by researchers in collaboration with a private company, which is experienced in developing serious games and e-learning. Parents have to register seven days (the previous week’s) of their child’s intake of discretionary food and drinks. A literature search showed that most previous screening tools aimed at children are food frequency questionnaires (FFQ) [35], and to our knowledge, no web-based screening tool to measure intake of discretionary food and drinks has previously been developed for individual use. The tool was developed through the following process: Firstly, the concept and aim were developed through a workshop with health practitioners, researchers and technical developers, and secondly, two different types of prompts were tested using a questionnaire on 18 parents. One type of registration had to be registered forward-prompting, starting Monday morning until Sunday evening, registering one day at a time. The other type of registration was prompting on meals and situations, where parents had the opportunity to register several days at a time, e.g., snack bar several weekdays in the lunch pack. The latter had better ratings among parents and proved more time efficient. Previous literature has shown that prompt types, either direct on meals or open prompts did not differ in improving accuracy, and that the efficacy of prompting on time, forward-prompting or backward-prompting, depended on sex [36]. Thirdly, the final tool was tested among parents and tool refinements were made. An advantage of the web-based tool is that it is very flexible, open, and non-linear.

#### 2.3.3. Game, App and Posters

A card-game and learning app were developed in collaboration with a private company experienced in developing serious games and e-learning. Conceptualisation of the card-game (Figure 4) was developed through a session with study researchers and health professionals (health nurses and dentists). Some key decisions were that the game needed to have short game sessions and be re-playable, include competition, involve the entire family and encourage dialogue about healthy and unhealthy dietary habits in a friendly tone. Through a concept and design phase, the game was tested through approximately 20 game sessions with the target group to refine actions, number of cards, winnings, etc. Through a production phase, graphics and a manual were further refined and tested in a target group. A cartoon monster for the game was created with the aim of externalizing unhealthy habits; the game is therefore called “The Monster Game”. The card game can be played in two different ways. First, as a regular competitive game, evolving around a monster where pictures of healthy food cards, sugar-rich food cards, mood cards, and situation cards, cause actions. Second, the cards can be used in a non-competitive game, were the aim is to dialogue when family members consume sugar-rich discretionary food and drinks.

A learning app (Figure 5) to be used on smart phones, is centred around the Monster, to create a coherent universe spanning the project as a whole. The Monster was designed through a process that involved mood boards and testing of sketches with the target group, children aged 5 to 10 years. The conceptualisation of the app involved nutrition experts from the research team and health professionals. There are two games available on the app. First, the Monster’s teeth can be fed with sugar-rich food and SSBs to see how they are affected. Further, the tummy of the Monster can be fed with sugar-rich food, replacing the healthy food. Secondly, an augmented reality (AR) function using monster stickers can “wake” monsters. Short messages can be recorded by the child and played back in a distorted funny voice. Testing of the Monster design was carried out with four parents and eight children. Testing of the app’s usability and likeness involved another four parents and eight children, and showed that the children enjoyed playing, particularly the AR function and tooth functionalities.

Posters (Figure 6) with the Monster were developed to be used at the sites of the school health nurses and dentists, but were also printed in smaller sizes to be delivered as part of the box for the intervention families. The poster evolves around the Monster, with quick response (QR) codes to download the app, and is aimed at both having a function as a conversation tool at the dentist and school health nurse, and increasing awareness of the project as a whole.

#### 2.3.4. Inspiration Booklet: Ideas for Changing Family Habits on Sugar-Rich Discretionary Food

The inspiration booklet was developed to cover a wide range of family practices and parenting styles gathered through a systematic combination of steps:

Step 1: In a targeted influencer campaign aimed at families with young children, four Danish influencers (Instagram users with an established audience) collected ideas, tips and tricks on reducing the amount of sugar-rich discretionary food among their followers on Instagram. The campaign resulted in 543 ideas. Identical ideas and duplets were excluded and 187 relevant ideas were selected. Based on content analysis and central concepts from the theoretical framework of SCT, ideas were systematised, condensed and categorised into the following thematic categories: monitoring (knowledge and expectations), limiting serving sizes, replacements (structural and social impediments), teaching (observational learning and knowledge), family rules and structures (family self-efficacy), limiting availability and accessibility (reciprocal determinism), and alternative activities (self-efficacy). Ideas were unevenly distributed between the categories as families had contributed with numerous ideas on family rules and structures (*n* = 53), replacement with healthier snacks (*n* = 51) and alternative activities (*n* = 31), whereas other categories had very few contributions e.g., limiting serving sizes (*n* = 10) or monitoring (*n* = 2). These insights informed the subsequent development of the booklet, as additional ideas and strategies were added in these sparsely represented categories to promote novel strategies and knowledge to the families to facilitate behaviour change.

Step 2: A workshop was conducted among external experts with different expertise in nutrition, food psychology, and teaching, where the selected ideas were reduced to 28 main ideas. The aim was both to select the most promising ideas, but also to refine and improve them. Next, 10 qualitative interviews on family habits and practices around sugar-rich discretionary food and drinks were conducted with parents from the target group in order to secure the relevance and usability of the selected ideas.

Step 3: The identified salient themes and ideas were summarised into short informative texts with checklists, fact boxes and illustrations to help the families change their habits on discretionary food and drinks in a collaborative process with a communication agency.

#### 2.3.5. Serving Size Board with Reusable Stickers

To illustrate the main messages, a small reusable sticker pad illustrating the maximum recommended amount of servings of discretionary food for 4–6 year-olds on one side and 7–9 year-olds on the other side, and 30 reusable stickers with different examples of 450 kJ servings of SSBs, cakes, confectionary, ice cream etc. The board can be placed on the refrigerator, and stickers can be added to monitor intake (Figure 7).

#### 2.3.6. Children’s Book

The box with materials also contains a copy of the children’s book “Anton og Sukkerdillen”. The focus of the book is to promote knowledge on dental health and to encourage a change in the habits of intake of sugar-rich discretionary food in a more sensible direction in 5–7-year olds and their families. The book is the result of a former collaboration between Danish dental organizations and has previously been tested in dental clinics in three Danish municipalities.

#### 2.3.7. Activity Suggestions

As inspiration for family activities, a special “municipality package” was compiled to be included in the box. The package includes folders, coupons for the local indoor swimming pool and tickets for two different family trips; one to the public library and a bus tour around the municipality, where the families can become acquainted with the leisure activities available for families with children.

#### 2.3.8. Communication and Social Media

Parents in the intervention group will be invited to join a closed Facebook group. The aim of the group is to provide information on positive dietary changes especially regarding the reduction in intake of discretionary food and drinks, increasing intake of water and healthy snacks including fruit and vegetables and engaging in more family activities. Further, it will encourage the use of the intervention home-use materials and facilitate interaction among participants. According to a recent review by Zarnowiecki et al. [37], content should be specific and relevant, such as serving sizes for different ages, and relevant for all family members, especially the child (e.g., appropriate recipes). Moreover, affirming content will be given, rather than negative content (e.g., avoiding a weight management focus or shaming).

The Facebook group will be moderated by a communication agency, with involvement of the research team. The Facebook group content will include approximately three posts per week during the entire intervention period including posts targeting one or more constructs of the SCT. Posts will include main themes provided in the inspiration booklet, child-friendly food ideas/recipes, motivational prompts on the use of intervention components (e.g., the Monster Game and the serving size board), practical tips and information (e.g., ideas for healthy snacks), and resources and opportunities related to the local area (e.g., local playgrounds and library). Each post will include a picture or other visual content. Interactivity will be supported by regularly including engaging and interactive components (i.e., videos, opinion polls, competitions and encouragement to share experiences). For parents that do not have a Facebook account, or do not want to join the Facebook group, emails with ideas and highlights from the group will be sent to the participants. There is no requirement for degree of involvement in the Facebook group, and no further contact will be made for non-engaged parents.

#### 2.3.9. Focus Group Evaluation of Intervention Materials

Evaluation of the different intervention materials and tools during the development process was performed by an external analytic agency that conducted two consumer focus groups. The purpose was to learn about the parents’ assessment of and opinions on a selected range of the intervention materials and tools. Two focus groups were conducted: One with participants of high-resource background (*n* = 6) and one with participants with low-resource background (*n* = 8). The key findings from the focus groups guided the finalising of the intervention materials.

### 2.4. Outcomes

#### 2.4.1. Primary Outcomes

The primary outcomes are changes in intake of pre-defined discretionary food and drinks measured by dietary assessment at baseline and a 3.5-month follow-up. Discretionary food are categorised as described in Section 2.3.1, and daily consumption are summarised for each participant.

Dietary assessment will be carried out using a web-based, 7-day dietary record. Parents are instructed at the baseline interview to fill out the dietary record on behalf of their child. A similar web-based dietary assessment software has been validated for children 8–11 years [38]. The dietary assessment software is structured by six eating occasions a day, breakfast, lunch and dinner, and three in-between meals after breakfast, after lunch and after dinner. For each meal, participants can search for food items, and choose between portion sizes among 1–4 different pictures for each item. If food items cannot be found, there is an option to add an open answer. Further, participants are reminded to register related food items such as spread (for bread) or milk (on cereals) and they are separately reminded to register drinks. For each meal, they are asked questions about where they had the meal, and whom they ate together with. Finally, participants are asked if they forgot to register sweets or chocolate, if they had any supplements, and if the day represented a usual or an unusual day and reasons such as birthday or illness. If parents fail to register for a day, they will be reminded by e-mail the following day, and receive a text message after two days of missing registrations.

Intake of food items, energy and nutrients are calculated for each study participant for each meal and as an average intake per day using the software system General Intake Estimation System (GIES) version 1.000.i6 and the Danish Food Composition Databank version 7.0, both developed at the National Food Institute, Technical University of Denmark. Mis-reporters of dietary intake are identified by evaluating reported energy intake using the Goldberg cut-offs for the ration between reported energy intake and estimated basal metabolic rate at individual level (BMR) as suggested by Black [39]. BMR will be calculated using gender-specific equations [40,41] with use of weight and height measured by school health nurses. At least four days including one weekend day have to be completed for an individual to be included in the analysis.

#### 2.4.2. Secondary Outcomes

Secondary outcomes are changes in intake of selected nutrients and food groups. Changes in added sugar (E %) will be evaluated in order to compare with international recommendations and other studies. Changes in energy intake as well as whole-grain products, dairy products, nuts, vegetables and fruits, saturated fat, and dietary fibre will also be evaluated. First, part of the advice during the intervention focuses on replacing unhealthy snacks with healthier ones, such as fruit, crispbread or nuts. Secondly, discretionary food and drinks take up room for healthy food, and therefore it is relevant to analyse which food components have potential to change. Although there is a large focus on added sugar, changes in saturated fat are likewise possible. Lastly, changes in total energy are analysed because some of the intake of discretionary food might be consumed as excess energy. FFQs were included in the questionnaire at both baseline and follow-up to ensure secondary analysis in case of large data gaps on dietary registration, and to capture not frequently consumed food and drinks. The FFQs asked about sugar-rich food and drinks-, fruit-, vegetable- and nut consumption.

Measurements of weight, height and waist circumference are obtained by the school health nurse at the consultation at baseline and at 3.5-month follow-up. Weight and height is measured by the school health nurses equipment and their usual method. All weight scales have been checked by the research team, with standard weights, to ensure accuracy. Waist circumference is measured according to WHO guidelines with a non-elastic measurement tape, measuring the approximate midpoint between the lower margin of the last palpable rib and the top of the iliac crest. The waist circumference measure is performed two times and the average is reported. Weight, height and waist circumference are used to evaluate weight status and abdominal status, and changes in anthropometry from baseline to follow-up between groups.

The questionnaire includes items about family activities because the intervention has a large focus on family activities, and as previous studies have shown, family time often is related to intake of discretionary food and drinks [42]. As part of the intervention aims to promote family activities that do not involve having discretionary food and drinks, family activities are measured. Two items were developed to quantify child and family time spent together, and one question on the most frequent activity done together. These questions were inspired by a previous study [43]. Specific family activities were identified from qualitative family interviews carried out as part of the preparation for the intervention.

Parents are further asked to estimate the amount of their children’s physical activity in hours per week. Screen time (TV and computer time in hours per day), and sleep duration (hours per night) will be collected at baseline and follow up, to measure potential changes related to health behaviour during the intervention. The use of these items is derived from the questionnaire used in the Danish National Survey of Diet and Physical Activity 2020–2021.

#### 2.4.3. Mediators

As a SCT approach informed the design of the intervention, these constructs were used to measure changes in potential mediators, at both baseline and follow-up questionnaire. Potential changes from baseline to follow-up will help determine if the intervention proves successful. Inspired from previous research [44,45], a questionnaire was developed to capture changes in; parental knowledge, parental expectations, behavioural capability, parental self-efficacy, and social/structural impediments. The SCT approach was combined with theory on parental practices to capture the dimension of parental intentions (i.e., expectations) and structures (i.e., social/structural impediments) [33]. An overview of the mediators measured in the questionnaire is provided in Table 2.

Two specific areas of knowledge were included. First, knowledge of the maximum intake advise for discretionary food and drinks choices in a healthy diet for children; and second, knowledge of official recommendations on fruit and vegetables for both children and adults. Items for knowledge were adopted from key messages used in the intervention “Are you too Sweet?” Because new guidelines and messages regarding discretionary food and drinks has been developed, potential changes in knowledge of these messages as well as what discretionary food can be replaced with is aimed for.

To measure baseline expectations and intentions, parents are asked both whether they believe their children consume too much discretionary food or drinks, and if they wish to change these or other dietary habits. Built on previous findings, the hypothesis is that parents do not believe their children consume too much, even though that might be the case [18].

According to Bandura, self-efficacy is key to behavioural change [33]. Measuring parental self-efficacy in relation to child feeding and snacking has been used in several interventions [47,48]. Because the intervention has a high degree of parental and overall family involvement, self-efficacy scales were divided into three domains. First, items from a validated Swedish survey instrument was used to measure overall self-efficacy regarding food and nutrition [48]. Second, a qualitative study about Danish parents’ perceived understanding and behaviour in relation to sugar-rich discretionary food and drinks [18] was used to address structural and social impediments [33]. These obstacles were included in the instrument to measure self-efficacy related to the habit of eating sugar-rich discretionary food in families. Third, as the intervention evolves around family interaction and activities, items of perceived collective family efficacy inspired from Bandura [49] were included.

Several items regarding parenting practices have been developed and tested previously [34,46,50,51]. These evolve around measuring degrees of *coercive control*, *structure*, *autonomy support*, and *permissiveness* [34]. These measures are used to describe expressions of modelling, structures and environment, and thus behavioural capability and impediments. Further, as the aim is to change children’s habits mediated by parents, it is important to capture which parental behaviours might be changeable. Eight items from the Comprehensive Snack Parenting Questionnaire (CSPQ) [46] were used after translation into Danish. Two independent translators, one nutritionist, and one non-specialist carried out the translation. Afterwards, translations were compared and agreed upon.

Based on previous Danish qualitative research [42], questions regarding norms and values related to discretionary food and drinks were developed. Previous research has shown that impediments for changing food habits often is found within both structural and social norms [18,42].

#### 2.4.4. Covariates

Demographic information on parental educational level, ethnicity, and household constitution is assessed through the questionnaire at the baseline interview. Both anthropometrics, ethnicity, and parental educational level, are compared with data from Hvidovre municipality to assess generalisability.

#### 2.4.5. Development and Validation of the Questionnaire

Previously validated questionnaires were used as described when relevant. The questionnaire was discussed in the research group to ensure coverage of relevant disciplines. One external expert on family life styles further revised questions on family activities. Face validity was tested with six parents in individual online think-aloud interviews in a pilot test. Think-aloud interviews were used to ensure that the questions were understandable and understood in the same manner across interviews. For example, the interviewer asked questions such as, *how do you understand this question?* Questions with more than one interpretation were removed from the questionnaire or specified in a more detailed manner. Time consumption and feasibility were further tested in a field test with 26 parents with children aged 4–9 years.

### 2.5. Process-Evaluation

A mixed method approach will be used to evaluate compliance as well as the feasibility and acceptability of the intervention. Fetters et al. states the default expectation for conducting interventional studies should be a combination of qualitative and quantitative components [52]. The evaluation will help understand how and why the intervention, or parts of it, was effective or ineffective and whether the intervention holds the potential to be successful in a wider context.

Quantitative measures will include questions in the post-intervention questionnaire on the number and kinds of activities participants have knowledge of, have tested, and are satisfied with, e.g., how many times the family played the card game. Further, participants will be asked to evaluate the usability of the main messages of the project. Open-ended questions will be included where participants are invited to voice their opinions and experiences. Finally, participants will be asked whether they would recommend the use of the various intervention components to others. With regard to the participants in the control group, they will receive additional questions related only to the health nurse consultation. Regarding the evaluation of the use of a social media platform (Facebook), different metrics will be used to measure volume of engagement, i.e., number of “likes”, total comments, and comments on sharing experiences. Engagement will be evaluated separately for different categories of posts.

Semi-structured qualitative interviews will be conducted post-intervention with 20 families from the intervention group in order to get a more in-depth evaluation of the intervention components and a better understanding of e.g., the motivators and barriers, compliance and food and activity habit changes made by the families. In addition, 2–3 focus groups will be conducted among the school health nurses having had consultations with either intervention or control families to evaluate intervention delivery at the consultation, suggestions for intervention component improvements and their views on future perspectives and possibilities for disseminating nation-wide. All interviews will be transcribed and coded into themes by the interviewer using the framework of content analysis [53].

Hvidovre municipality is affected by the COVID-19 epidemic with several school classes closing for one or several weeks with short notice. Information on infections and closed school classes will be collected as dietary habits and compliance to the intervention might be affected by the situation.

### 2.6. Pilot Study

A pilot study was carried out to assess measurement procedures, questionnaires and other data collection instruments, logistics and suitability of the intervention components for the target age group. Due to the Covid-19 context, the pilot study had to (1) be postponed from May 2020 to August 2020 and (2) reduced from six weeks to four weeks. The compressed procedure taken into account the follow-up measurements were omitted. Eight 1st grade children and their parents participated in the pilot study. All eight families completed all intervention components, but the measurement of pre-post differences. The test thus comprised an evaluation of the questionnaire, the 7-day dietary record, the sugar-rich food screener, the consultation with school health nurse, the take-home box as well as a communication and social media platform. Materials were well received among both school health nurses, parents, and children following minor adjustments, primarily related to the design and phrasing in the inspiration booklet, and rules in the game to be easier to understand. Parents stressed how the consultation and the material from the take-home box jointly and in synergy provided a vehicle for behaviour change. Consequently, the pilot study provided useful and relevant data on practicality, sensitivity, and suitability of materials.

### 2.7. Statistical Considerations

Primary analysis will focus on testing the effectiveness of the intervention, by analysing the change in intake discretionary food and drinks from baseline to the 3.5-month follow-up between intervention and control groups. Changes in intake of discretionary foods and drinks will be expressed in absolute changes (kJ/d) and in relative changes as kJ/10 MJ. Changes in food groups and sub-groups (e.g., chocolate, cakes or ice-cream), will also be evaluated in g/d.

Linear mixed models will be used with discretionary food and drinks as a combined variable, and sugar-rich discretionary food and discretionary drinks separately as outcome variables. Group (intervention or control) baseline intake of discretionary food and discretionary drinks will be analysed as a fixed effect and school as a random effect as there has been a cluster randomisation on school level. Primary outcome will be calculated using intention-to-treat analysis, including all participants irrespective of adherence or dropouts. Missing data will be imputed using multiple imputation. Sensitivity analysis will be performed with complete case analysis, and per protocol analysis, where analysis will be carried out for participants that (1) complete the sugar-rich food screener and health consultation; (2) report to play the card game and used at least one other item from the box. Further, sensitivity analysis will likewise be used to test the effect of having twins or triplets in the intervention.

Adjustment for group imbalances and potential confounders such as parental education, BMI and physical activity will be performed for both primary and secondary outcomes. Estimates of the effect size will be presented along with 95% confidence intervals and p-values. The significance level is set at 5%. Statistical analysis will be carried out using R version 4.0.

### 2.8. Consent to Participate and Ethics

As the intervention is non-invasive, the local ethics committee confirmed that no official approval is needed (journal-nr.: H-20036402). The study is conducted in accordance with the Declaration of Helsinki and written informed consent of parents has been obtained before participation.

## 3. Discussion

Collaboration with the local school health nurses and child dentists could provide an effective structural frame to support behaviour change and anchor new initiatives. However, evidence-based prevention strategies in practice within school health services are still limited [54]. In this multi-component theory-based intervention the effectiveness of disseminating new guidelines on sugar-rich discretionary food and drinks through local school health nurses will be evaluated. The intervention and its wide range of materials will contribute with new insights to this area. The intervention “Are you too Sweet?” has been developed in collaboration with both school health nurses and dentists to suit current practices, thereby ensuring applicability and the likelihood of sustained health-promotion. Further, communication strategies have been developed together with communication partners and tested both through focus groups and in a pilot study to ensure relevance for parents in the target-group.

A high consumption of sugar-rich discretionary food and drinks, especially SSBs, has several health implications, tracking from childhood to adulthood. Prevention of overweight, dental caries and a nutritionally poor diet are concerns that need to be addressed in early school years. Parents as caregivers are the primary role models in shaping their children’s eating habits, and thus, an essential part of the intervention. In recent years, several concepts regarding parenting practices have been developed in relation to family nutritional practices [34,46,50,51]. Systematic reviews have found that control reflected as pressure and restrictiveness have been associated with a higher intake of discretionary food, while parental monitoring and limited home access to discretionary food, have been associated with a lower intake of discretionary food [17,34]. The importance of both accessibility and availability, e.g., as children may not consume fruits and vegetables that are inaccessible, even if available, especially if unhealthy food are also available or accessible, has recently been highlighted [55]. Further, parents acting as positive role models have been associated with lower consumption of SSBs among children, whereas using food as a reward was associated with a higher intake [56]. Qualitative studies have shown that parents face difficulties controlling their children’s intake of sugar-rich discretionary food [57], but also that family eating habits are decisive for the level of intake of sugar-rich food and SSBs [18]. “Are you too Sweet?” encourages parents to both act as role models, as well as limit the availability of sugar-rich food, and make healthy snacks accessible. However, the intervention is limited from structural changes, such as increase in certain tax levels that more directly could enhance changes in accessibility [58].

A strength of this intervention is that the school health nurses are highly experienced in anchoring health initiatives and engaged with parents in shaping healthy habits. Further, joining multiple output data on food intake, anthropometry and a range of behaviour change mediators is a strength of our study design, as hypothetical mechanisms of behavioural change will be discussed. On the contrary, analysis of each mediator might be constrained due to low power. A rather large dropout due to delay caused by the novel COVID-19 virus, has weakened the power, resulting in a change to the study design. Therefore, the randomisation has not been optimal, as one school had to be changed from control school to intervention school [29], and that randomisation had to be performed before baseline measures. The COVID-19 situation might influence peoples eating habits as they stay more at home, and school afternoon snacks are not available, but have to be bought from home. The schools included in this project, do not plan to participate in other health related projects related to the COVID-19 situation.

Further, the allocation status could not be blinded, for obvious practical reasons, for participants or those delivering the interventions. However, randomisation was blinded to the participants until after they completed a 7-day dietary record at baseline. In contrast, a strength is that using current school health services has facilitated that families with different socio-economic backgrounds have been enrolled in the intervention. That said, the representativeness of the study might be reduced because some parents declining to participate, as they reported a lack of sufficient resources to fill out the dietary record and survey. The representativeness will therefore be compared with local data of education and ethnicity when data collection is finished.

The pilot study indicated that the measurement procedures and logistics of the intervention is feasible. However, the pilot study had several weaknesses. Some families had too little time to use and test all materials provided by the project due to the short period. The intervention therefore commenced with only minor changes.

This paper has thoroughly described the development of all components used in the intervention in order to ensure a critical evaluation of all components afterwards. Further, statistical considerations have been described in order to avoid data driven analysis.

## 4. Conclusions

The trial “Are you too Sweet?” will test the effectiveness and feasibility of communicating new Danish guidelines for discretionary food and drinks for school starters (5–7 years) through local school health service. New materials and tools were developed to measure and promote changes. This includes extended dialogue at the school health nurses and a take-home box for participants as well as a communication and social media platform to facilitate interaction among participants. The intervention is ongoing and 160 children and their parents are scheduled for a baseline interview at six different schools. This study will add knowledge to the efficacy of a multicomponent intervention directed at families to increase their self-efficacy and knowledge and thereby instigate behavioural change. In a Danish context the concept of “hygge” implies cosy family time, especially in the weekends where confectionary, SSBs, snacks, desserts and cakes play an important role, and the intake of sugar-rich discretionary food and drinks thus has an important cultural and traditional meaning [18,42]. Although, cultural changes and social impediments might be difficult to change at a family level, “Are you too Sweet?” will inspire to family time not necessarily including sugar-rich discretionary food and drinks. If proven effective, wide-scale implementation in other municipalities, could lead to more effective primary prevention and a general lower consumption of sugar-rich discretionary food and drinks in the population.

## Figures and Tables

**Figure 1 ijerph-17-09580-f001:**
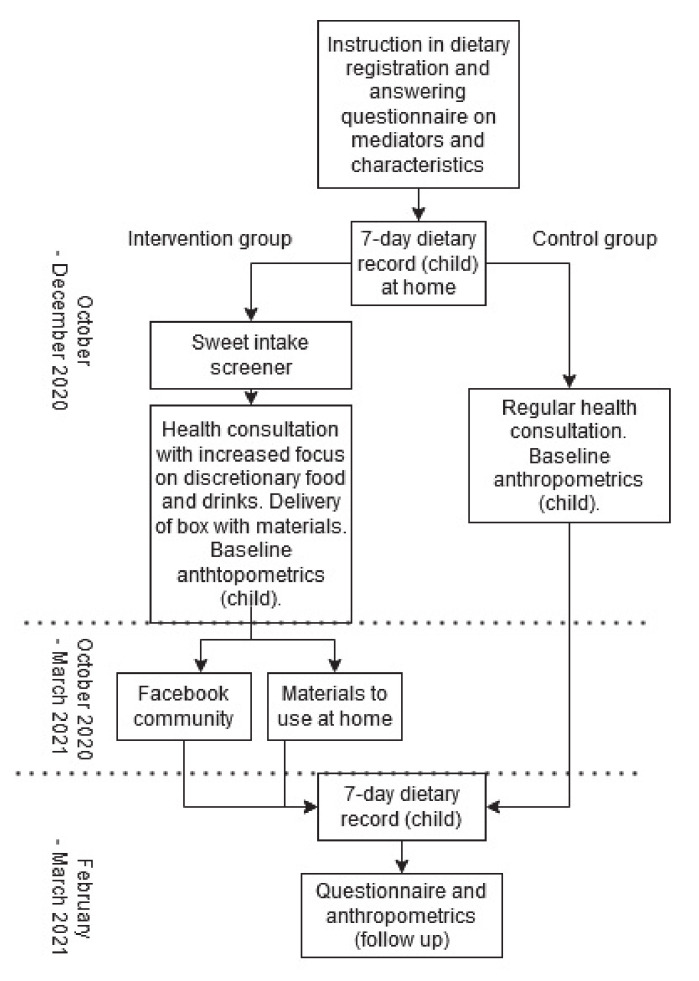
Study flow of the “Are you too Sweet?” intervention.

**Figure 2 ijerph-17-09580-f002:**
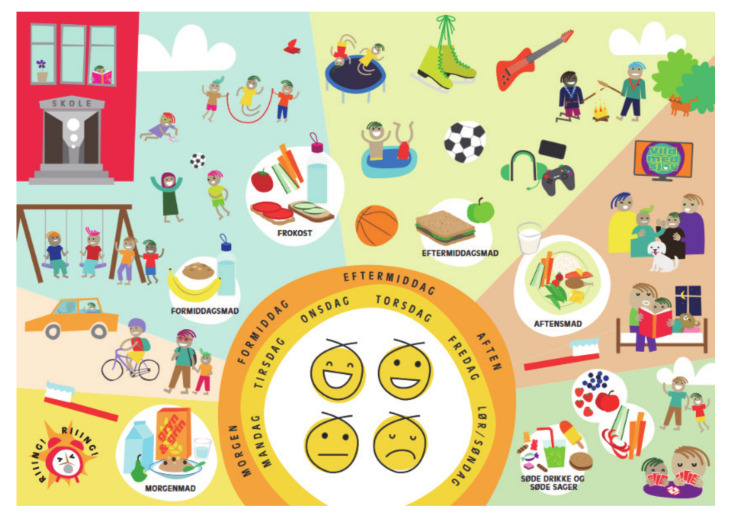
Conversation tool for school health nurse.

**Figure 3 ijerph-17-09580-f003:**
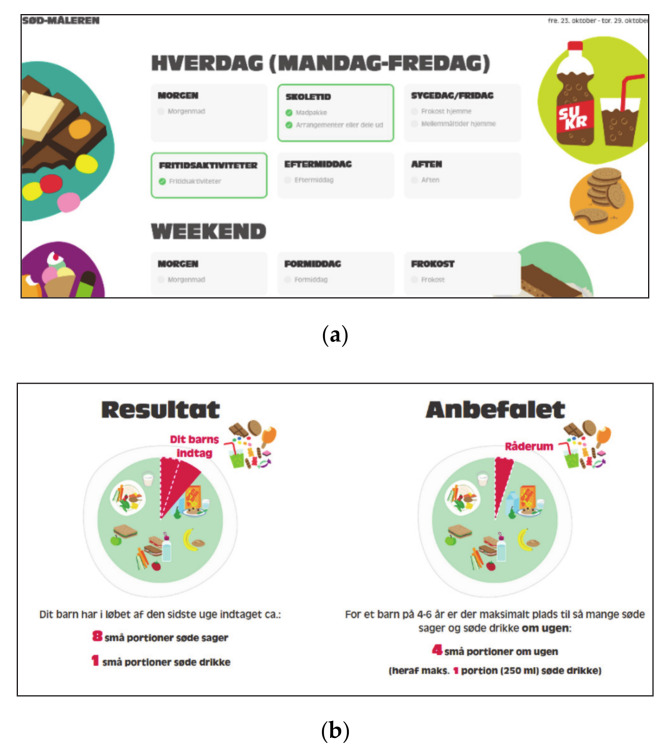
Screenshots of the sugar-rich food screener. It assesses one week’s intake of discretionary food and is completed in approximately 10 min. (**a**) The sugar-rich food screener is non-linear and can be completed in the order participants prefer. (**b**) When completed, the last page illustrates how many serving sizes the child has consumed, and the share that the sugar-rich food and drinks take up from other food compared to recommended.

**Figure 4 ijerph-17-09580-f004:**
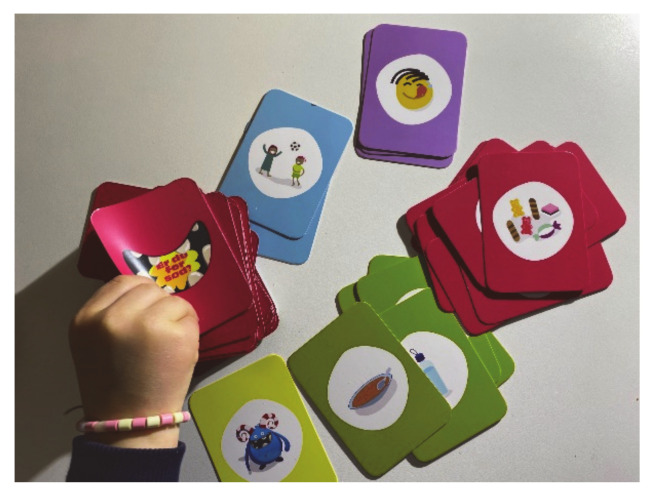
The card game can be played in two ways, but can also be played with as the child wishes. The cards illustrate healthy foods, discretionary foods, mood cards, situation cards and a monster.

**Figure 5 ijerph-17-09580-f005:**
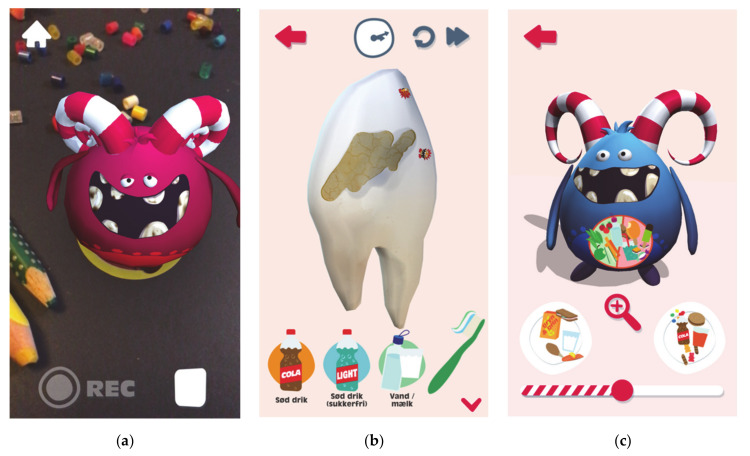
Features of learning app. (**a**) Augmented reality (AR) function wakes a monster when an AR sticker is scanned using a smartphones rear camera; (**b**) a tooth can be exposed to both discretionary drinks and different food, but you have to brush right away to prevent damage; (**c**) the Monster can be fed with both discretionary food or healthy food, but there is not room for both.

**Figure 6 ijerph-17-09580-f006:**
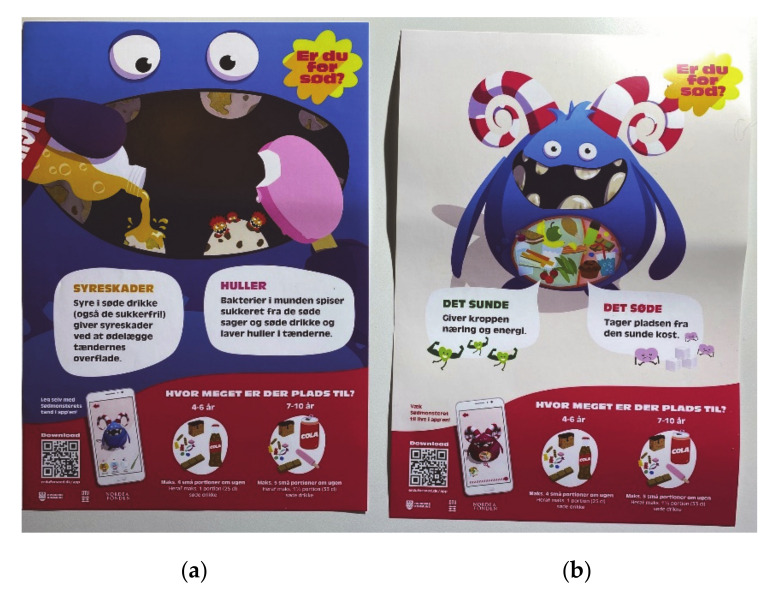
Posters are both part of the box given to families, and developed for consultation rooms for (**a**) the dentists; (**b**) school health nurses.

**Figure 7 ijerph-17-09580-f007:**
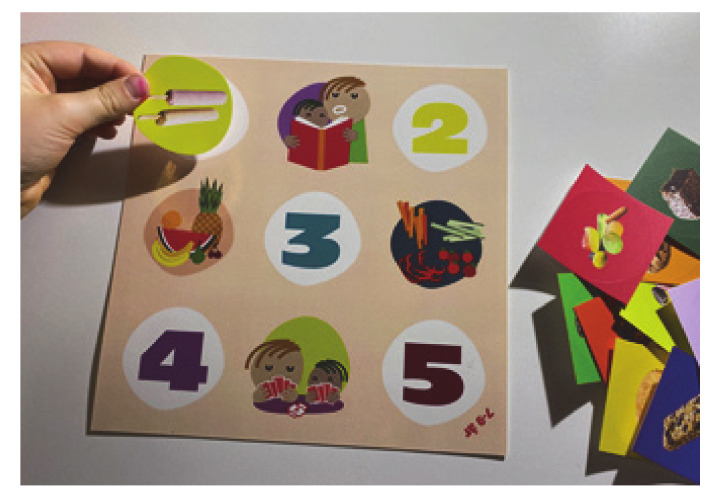
Serving size board. Here are illustrated five servings, but the stickers can easily be removed and replaced with others the week after.

**Table 1 ijerph-17-09580-t001:** Overview of intervention components in relation to the theoretical construct. Different colour shades illustrate main elements of the intervention: New guidelines; assessment and consultation; Home-use materials; and Communication and social media platform.

Activity	Theoretical Construct of Behaviour Change
New guidelines The amount of energy from discretionary food and drinks choices in a healthy diet, communicated as a maximum amount of servings.	Foundation of intervention to boost *knowledge* on how much discretionary food and drinks are compatible with a healthy diet.
Sugar-rich food screener Online tool to register the child’s intake of sugar-rich discretionary food and drinks.	Increased awareness and *knowledge* of current behaviour, followed by a qualification of the school health nurse consultation with increased focus on *expectations.*
School health nurse consultation Extended consultation with increased focus on discretionary food and drinks.	Structural frame to anchor, motivate and communicate guidelines and deliver materials.
Game Card game, posters, and app, externalizing bad habits through the “Monster”.	Increasing *self*-*efficacy* through game and fun activities. The game and monster figure will effect *expectations* through increased awareness.
Inspiration booklet Concrete advice to choose from, how to implement new habits, guides to healthy snacks, awareness of parental role.	Increases *knowledge* and *behavioural capability*. Parental awareness of their own role may affect *observational learning.*
Children’s book About a crocodile, that loves sweets, but gets dental problems.	Increasing children’s and parents’ knowledge on dental health*Observational learning*.
Activity suggestions Coupons for the local indoor swimming pool, pamphlets about local playgrounds and other facilities, invitations for a library guided tour, and a bus tour.	Awareness towards offers and activities in the local community, to increase focus on family time as something physical and/or cultural, and promote family time as not related to food, and thus limit *social*/*structural impediments.*
Serving size board Board illustrating serving sizes for discretionary food and drinks.	Promotion of servings and helps to increase *behavioural capability*, as it is a concrete tool to measure a week’s intake of sugar-rich food.
Communication and social media platform Facebook community provide information and interaction between participating parents.	Promote community self-help and peer-to-peer support, through *reciprocal determinism* and *observational learning*. Further, to keep awareness and motivation of participation.

**Table 2 ijerph-17-09580-t002:** Overview of the theoretical construct of the questionnaire.

Theoretical Construct	Mediator in Questionnaire
Knowledge	Parents’ nutritional knowledge e.g., *“How much sugar-rich food is there room for in a healthy diet?”/“What are the recommendations on fruit and vegetables?”*
Expectations	Expectations for healthy eating e.g., *“Would you like your child to eat less sugar-rich food?”*
Behavioural capability	Parental intentions (based on questions from the *Comprehensive Snack Parenting Questionnaire* [46]) e.g., *“ I limit the availability of sugar-rich food in the house for [child’s* *name]”/“I give [child’s name] sugar-rich food to make [him/her] feel better.”* */“I consciously refrain from eating sugar-rich food when [child’s* *name] is around.”*
Social and structural impediments	
Observational learning	
Self-efficacy	Parents’ belief in his or her capacity to execute behaviours e.g., *“How confident are you, that you can offer meals with vegetables for [child’s name]?”/“I think it is difficult to obtain limits to [child’s name] intake of sugar-rich food or drinks, when my child is together with grandparents or other family”/“How confident are you, that your family can agree on shared goals?”*
Social and structural impediments	Parents’ norms and values e.g., *“Children can have sugar-rich drinks during weekdays, e.g., to have sweet drinks in a drinking bottle to quench their thirst”/“Children should not eat sugar-rich food every day.”*

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
