# Peer review of "Reducing Young Schoolchildren’s Intake of Sugar-Rich Food and Drinks: Study Protocol and Intervention Design for “Are You Too Sweet?” A Multicomponent 3.5-Month Cluster Randomised Family-Based Intervention Study"

_ijerph, 2020, doi:10.3390/ijerph17249580_

Round 1

Reviewer 1 Report

First of all, it has been a great effort from the authors to design this project and putting together this manuscript. Reducing intake of sugar-rich discretionary food and beverages is still an important target for public health interventions and thus, this study can add to finding the best ways forward.

I have, however, some considerations to improve the overall manuscript.

  • Lines 52-42, 151, 203 ("worlds", 207, 250 and 363 need revising as there are grammatical mistakes that hinder clarity.
  • Lines 114 needs a reference for statement "25 % reduction in added sugar intake is considered a success".
  • Table 1: the theoretical construct for the "game activity" needs grammatical revision.
  • Line 138, provide the number of schools that have been randomised to control and intervention (n=; n= ).
  • Figure 1: study flow should include the parental questionnaire that has been developed.
  • Line 487 (on statistical analysis) will you have a combined variable (with food and beverages) and these former variables separately (only food and only beverages) for linear mixed model analyses?. If so, then writing needs improvement.
  • How will the primary outcome be measured? kJ/day; kcal/day; g of sugar/day? This is very important to be portrayed.
  • Would you consider parental BMI as part of the sensitivity analyses?
  • Will any changes in children or parental knowledge, awareness, confidence or any other theoretical construct of behaviour change be measured? This may be important considering the stages of behaviour change throughout the study; therefore, it will help determining if the intervention is being successful on impacting short-term; medium-term and/or long-term objectives.
  • Considering the current COVID-19 context, are there any other interventions (from the government, NGOs, industry) that may interfere with your outcomes (e.g., health campaigns, marketing, etc)?
  • Finally, for parents that show little involvement on the facebook page, would there be any reinforcement or contact?

Reviewer 2 Report

In general the study is well designed and described. In addition to dealing with a topic of great interest. Some points that can be clarified are:

  • Sentence starting in line 41: The construction of the sentence make it hard to follow. Please, reformulate the sentence.
  • Sentence in line 53: The sentence is hard to follow because there are some missing commas. Please, check the sentence.
  • Lines 158-159: Are both groups (intervention and control) included in this part of the study? Please, clarify it in the text.
  • Section 2.3.8: Please, specify in the text if there was any researcher of the study or any healthcare worker moderating the Facebook group.
  • How did you clasify children in the different physical activity or sedentary behaviour levels? Please go a little deeper at this point.
  • Do you use any standarized method to assess the anthropometric meassures? Please describe how these meassures were taken. 

Reviewer 3 Report

Looking at the dates and the length of the intervention, the purpose of the manuscript is unclear given that the intervention is over or nearly over. I suggest that the authors write a manuscript conveying the results of the pilot study and the from the development of the tools that are being used. The development of the tools used in this study (screener, game, app) would be of interest to readers and I am looking forward to seeing how the children adapted the message(s) from these. 

Verb tenses throughout need to be reconciled  

Line 29 - 7-day

Line 43 - delete the space after 13

Line 50 - sentence needs revision, perhaps "The Nordic Nutrition Recommendations (NNR) also advises consuming a diet with less than 10 E% from added sugar" 

Line 51 - this sentences needs to be revise to clarify what the authors are trying to convey to the reader

Section 2.3.4 - Results from the development of the Inspiration booklet should be presented

Section 2.6 - What were the results from the pilot study

Round 2

Reviewer 3 Report

The manuscript is much improved from the original version. There are only minor revisions needed.

Line 52 - need to bring the recommendation of <10E% in alignment with the quote recommending <6E%

Line 163 - delete the last comma

Line 198 - delete the first two commas

Line 283 - delete e.g.

Line 313 - change 'for thebox' to 'to be included in the box'

Line 350 - add a space after drinks

Line 354 - delete previously

Line 376 - add a space after the period

Line 378 - this sentence needs to be reworded; suggestion - 'Changes in energy intake as well as whole-grain products, dairy products, nuts, vegetables and fruits, dietary fibre, and saturated fat will also be evaluated.'

Line 403 - you state previous studies but only cite a single study, this needs to be corrected

Line 512 - I believe the 2.7 Statistical consideration is meant to be on a separate line and not part of this paragraph

Author Response

Response to reviewer 3 comment's

Thank you very much for your thorough review. We have revised all your corrections.

Line 52 - need to bring the recommendation of <10E% in alignment with the quote recommending <6E%.

Thank you, we corrected the confusing formulation, as we meant it to be even lower than <10E%. But we agree that it sounds incorrect.

Line 163 - delete the last comma

Line 198 - delete the first two commas

Line 283 - delete e.g.

Line 313 - change 'for thebox' to 'to be included in the box'

Line 350 - add a space after drinks

Line 354 - delete previously

Line 376 - add a space after the period

Thank you for all the above comments and corrections. We corrected them all accordingly.

Line 378 - this sentence needs to be reworded; suggestion - 'Changes in energy intake as well as whole-grain products, dairy products, nuts, vegetables and fruits, dietary fibre, and saturated fat will also be evaluated.'

Thank you for your nice suggestion.

Line 403 - you state previous studies but only cite a single study, this needs to be corrected

Line 512 - I believe the 2.7 Statistical consideration is meant to be on a separate line and not part of this paragraph

Thank you for discovering these mistakes, we have corrected them accordingly.